# Characterization of β-Glycosidase from *Caldicellulosiruptor owensensis* and Its Application in the Production of Platycodin D from Balloon Flower Leaf

**Kyung-Chul Shin [1], Min-Ju Seo [1], Dae-Wook Kim [2], Soo-Jin Yeom [3,\*] and Yeong-Su Kim [2,\*]**

[1] Research Institute of Bioactive-Metabolome Network, Konkuk University, Seoul 05029, Korea; hidex2@naver.com (K.-C.S.); cocoa10073@naver.com (M.-J.S.)

[2] Forest Plant Industry Department, Baekdudaegan National Arboretum, Bonghwa 36209, Korea; dwking@bdna.or.kr

[3] School of Biological Sciences and Technology, Chonnam National University, Gwangju 61186, Korea

\* Correspondence: soojin258@chonnam.ac.kr (S.-J.Y.); yskim@bdna.or.kr (Y.-S.K.); Tel.: +82-62-530-1911 (S.-J.Y.); +82-54-679-2740 (Y.-S.K.)

**Abstract:** Platycodin D has diverse pharmacological activities. An efficient and economical mechanism for obtaining platycosides (platycodin D in particular) would be very useful. Balloon flower leaf extract (BFLE) was obtained by recycling leaves discarded from Platycodi radix production, as they have a high platycoside E content. A recombinant β-glycosidase from *Caldicellulosiruptor owensensis* was characterized and applied to BFLE for platycoside bioconversion. The enzyme specifically hydrolyzed the glucose residue at the C-3 position in platycosides and was suitable for platycodin D production. Under optimized reaction conditions, β-glycosidase from *C. owensensis* completely converted platycoside E from BFLE into platycodin D with the highest concentration and productivity reported so far. These results greatly improve the production process for deglycosylated platycosides.

**Keywords:** *Caldicellulosiruptor owensensis*; β-glycosidase; balloon flower leaf; platycoside; platycodin D

---

## 1. Introduction

Balloon flower (*Platycodon grandiflorum*) has been known as a health food and a conventional medicine for treating bronchitis, tuberculosis, asthma, diabetes and other inflammatory diseases in Northeast Asia. Over the last decade, interest in platycosides, which are balloon flower saponins, has increased because of their diverse pharmacological activities. Of these, platycodin D, a major platycoside in the root of *Platycodon grandiflorum* (Platycodi radix), has shown diverse pharmaceutical effects such as anti-tumor [1,2], anti-inflammatory [3,4], anti-allergy [5,6] and anti-obesity [7,8] activities.

Platycodin D is comprised of a pentacyclic triterpene aglycone and two-sided sugar components that contain one glucose molecule at the C-3 and the oligosaccharide residue consisting of arabinose-rhamnose-xylose-apiose at C-28. It can be converted by deglycosylation from platycoside E and platycodin D3 (Figure 1), which account for about 20 and 3% of the total platycosides in Platycodi radix and have two and one more glucose molecules at C-3, respectively [9].

Deglycosylation of saponins improves their biological activity because of their resulting smaller molecular weight, better permeability through the cell membrane and higher bioavailability [10,11]. Therefore, various methods have been tried for saponin deglycosylation, among which, enzymatic conversion displayed superior selectivity and the highest productivity. For platycoside deglycosylation,

not only β-glucosidases from *Aspergillus usamii* [12], *Aspergillus niger* [13], snailase [14], laminarinase [15], and cellulase [16], but thermophile-derived recombinant enzymes such as β-glucosidases from *Caldicellulsiruptor bescii* [17] and *Dictyoglomus turgidum* [18] have been used. Recombinant enzymes are industrially useful due to the high enzyme expression rate; those derived from thermophiles have a higher stability and activity that is advantageous for industrial production.

In this study, β-glycosidase from thermophilic bacterium *Caldicellulosiruptor owensensis* was cloned, characterized, and applied to produce platycodin D. For the economical production of platycodin D, balloon flower leaves discarded during Platycodi radix harvesting and containing a large amount of platycoside E were used as a substrate. Reaction conditions for the production of platycodin D from balloon flower leaf extract (BFLE) were optimized and, under these conditions, platycodin D was produced from BFLE with the highest productivity to date.

**Figure 1.** Deglycosylation pathway of platycoside E to platycodin D. Red represents the carbon number in pentacyclic triterpene aglycone.

## 2. Results and Discussion

### 2.1. Cloning of Gene, Purification of Enzyme, and Determination of Molecular Mass

The *C. owensensis* β-glycosidase was expressed in *Escherichia coli* BL 21 with a soluble form. The expressed β-glycosidase from *C. owensensis* was purified using HisTrap affinity chromatography into a soluble protein with a 17-fold of purification and a final yield of 18%. The specific activity of recombinant β-glycosidase from *C. owensensis* was 51.6 μmol/min/mg for pNP-β-D-glucopyranoside from crude enzyme extract.

The molecular mass of a recombinant β-glycosidase was approximately 53 kDa in SDS-PAGE (sodium dodecyl sulfate-polyacrylamide gel electrophoresis). The calculated molecular mass of recombinant β-glycosidase was 54,018 Da based on 458 amino acids containing six histidine residues (Figure 2a). The native β-glycosidase from *C. owensensis* was determined to be a homodimer with 106 kDa of molecular mass by gel-filtration chromatography (Figure 2b). No peak other than β-glycosidase was detected in the chromatography (data not shown).

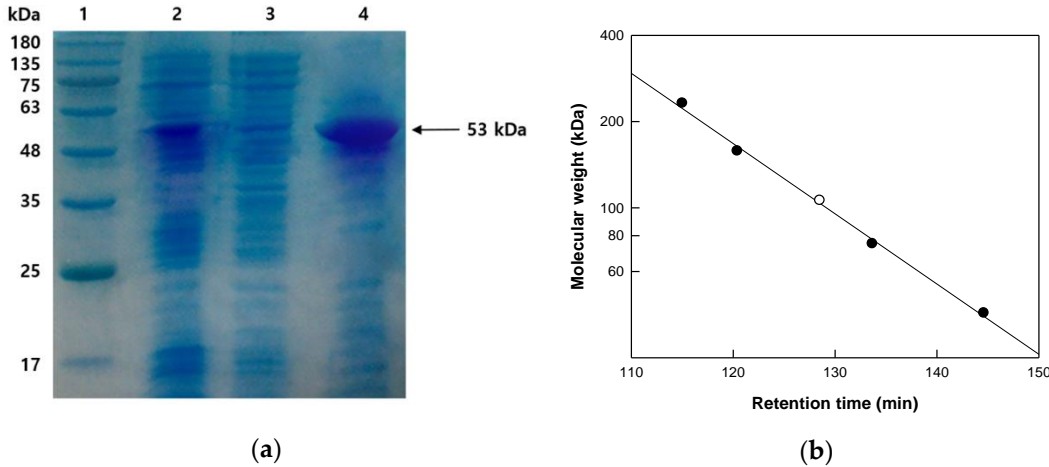

(**a**)                                          (**b**)

**Figure 2.** SDS-PAGE and gel-filtration chromatography of β-glycosidase from *C. owensensis*. (**a**) Analysis of SDS-PAGE for β-glycosidase from *C. owensensis*. Lane 1, molecular weight size marker; lane 2, cellular debris; lane 3, crude enzyme extract; and lane 4, purified β-glycosidase from *C. owensensis*. (**b**) Determination of molecular mass for β-glycosidase from *C. owensensis* using gel-filtration chromatography. The reference proteins and purified β-glycosidase from *C. owensensis* are represented by filled circles and open circle, respectively.

## 2.2. Hydrolytic Activity According to pH and Temperature Change

The recombinant β-glycosidase from *C. owensensis* showed maximum hydrolytic activity at pH 5.0 and 80 °C (Figure 3). Optimum pH and temperature of recombinant enzymes used for platycoside deglycosylation, β-glucosidases from *Caldicellulosiruptor bescii* [17] and *Dictyoglomus turgidum* [18] were pH 5.5 and 80 °C, and pH 6.5 and 80 °C, respectively.

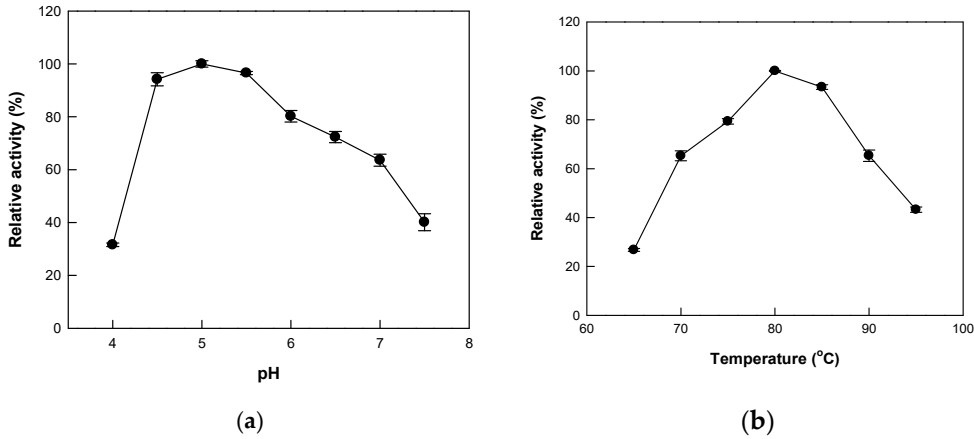

(**a**)                                          (**b**)

**Figure 3.** The activity of β-glycosidase from *C. owensensis* with changes in pH and temperature. (**a**) Effect of pH. (**b**) Effect of temperature. Data are represented in the means of triplicate experiments, and error bars are shown by the standard deviation.

The thermostability of β-glycosidase from *C. owensensis* was examined with the temperature range of 70 to 90 °C. First-order kinetics was displayed for thermal inactivation by the enzyme and the half-lives of β-glycosidase from *C. owensensis* were 107, 32, 12.5, 0.6 and 0.1 h at 70, 75, 80, 85 and 90 °C, respectively (Figure 4). Platycodin D-producing enzyme β-glucosidase from *C. bescii* displayed half-lives of 96, 29, 6.2, 0.1 and 0.03 h at 70, 75, 80, 85 and 90 °C, respectively. β-Glycosidase from *C. owensensis* had higher thermal stability than β-glucosidase from *C. bescii* across the board and β-glycosidase from *C. owensensis* showed about 2-, 6- and 3-fold higher stability at 80, 85 and 90 °C, respectively.

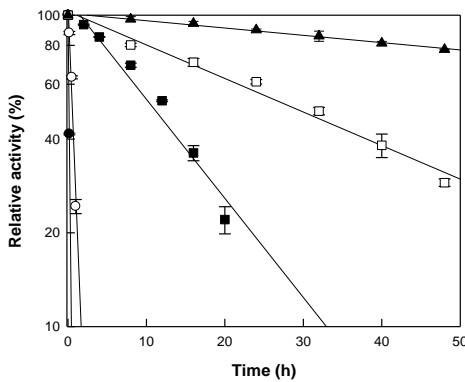

**Figure 4.** Thermal inactivation of β-glycosidase from *C. owensensis*. The enzyme was placed at 70 (closed triangle), 75 (open square), 80 (closed square), 85 (open circle) and 90 °C (closed circle) in 50 mM CPB (pH 5.0) for different time periods and then withdrawn at each time. Data are represented as means of triplicate experiments and the standard deviation was indicated by error bars.

## 2.3. Substrate Specificity

The substrate specificity of recombinant β-glycosidase from *C. owensensis* was measured using aryl-glycosides and platycosides (Table 1). The enzyme showed the following specific activity with aryl-glycosides as substrates: pNP-β-ᴅ-glucopyranoside > oNP-β-ᴅ-glucopyranoside > pNP-β-ᴅ-galactopyranoside > oNP-β-ᴅ-galactopyranoside > pNP-α-ʟ-arabinopyranoside > pNP-α-ʟ-rhamnopyranoside > oNP-β-ᴅ-xylopyranoside > pNP-β-ᴅ-xylopyranoside. The enzyme acted on diverse linkages in different glycosides, such as arabinopyranoside, rhamnopyranoside and xylopyranoside, as well as glucopyranoside and galactopyranoside. Nevertheless, the enzyme did not cleave the residues of arabinose, rhamnose and xylose at C-28. Based on the deglycosylation pathway of platycoside E to platycodin D, the enzyme is an exo-type hydrolase that hydrolyzes from an externally located sugar in turn. Since the enzyme did not hydrolyze the outermost linked apiose at C-28, the other sugars linked inside at C-28 such as arabinose, rhamnose and xylose could not be hydrolyzed.

**Table 1.** Substrate specificity of the β-glycosidase from *C. owensensis*.

| Substrates | | Specific Activity (μmol/min/mg) |
|---|---|---|
| Aryl-glycoside | oNP-β-ᴅ-glucopyranoside | 46.0 |
| | oNP-β-ᴅ-galactopyranoside | 21.0 |
| | oNP-β-ᴅ-xylopyranoside | 6.6 |
| | pNP-β-ᴅ-glucopyranoside | 51.6 |
| | pNP-β-ᴅ-galactopyranoside | 35.7 |
| | pNP-β-ᴅ-xylopyranoside | 5.5 |
| | pNP-α-ʟ-arabinopyranoside | 11.5 |
| | pNP-α-ʟ-rhamnopyranoside | 8.6 |
| Platycoside | Platycoside E | 72.5 |
| | Deapi-platycoside E | 60.6 |
| | Platycodin D3 | 2.5 |
| | Deapi-platycodin D3 | 1.1 |
| | Platycodin D | ND |
| | Deapi-platycodin D | ND |

ND, not detected.

The specific enzyme activity with platycosides as substrates showed the following order: platycoside E > deapiosylated(deapi-) platycoside E > platycodin D3 > deapi-platycodin D3. Platycoside E, deapi-platycoside E, platycodin D3 and deapi-platycodin D3 were converted to platycodin D3, deapi-platycodin D3, platycodin D and deapi-platycodin D, respectively. However, there was no

activity for platycodin D and deapi-platycodin D, indicating that the enzyme could not catalyze the inner glucose cleavage at C-3 position of platycosides and had significantly higher activity against the outermost glucose. The activities of β-glycosidase from *C. owensensis* for platycoside E, platycodin D3 and deapi-platycodin D3 were 1.06-, 6.25- and 5.5-fold higher, respectively, than those of a platycodin D-producing enzyme (β-glucosidase from *C. bescii*) [17]. However, β-glucosidase from *C. bescii* represented higher specific activity for deapi-platycoside E as a substrate than β-glycosidase from *C. owensensis*.

## 2.4. Optimization of Reaction Conditions

BFLE was obtained with extraction in 80% (*v/v*) methanol and the extract contained 5.19 mg/mL platycoside E, that constituted 92.31% (*w/w*) of the total platycosides (Table 2). These were 3.8- and 2.3-fold higher values than the concentration and content of platycoside E, respectively, in Platycodi radix extract (PRE) used for platycodin D production [9,19]. However, platycosides such as platycodin D3, polygalacin D3, deapi-platycodin D3 and 3'-*O*-acetyl polygalacin D3 contained in PRE were absent in BFLE. The content of specific platycosides in the total BFLE platycosides followed the order: platycoside E (92.31% (*w/w*)) > deapi-platycoside E (2.89%) > polygalacin D (2.12%) > platycodin D (1.34%) > deapi-platycodin D (0.53%) > 3''-*O*-acetyl polygalacin D (0.45%) > platycodin A (0.36%), indicating that BFLE is an efficient substrate for platycodin D production.

**Table 2.** Platycoside content in 10% (*w/v*) BFLE.

| Platycoside | Content (%, *w/w*) | Concentration (mg/mL) |
|---|---|---|
| Deapi-platycoside E | 2.89 | 0.16 |
| Platycoside E | 92.31 | 5.19 |
| Deapi-platycodin D | 0.53 | 0.03 |
| Platycodin D | 1.34 | 0.07 |
| Polygalacin D | 2.12 | 0.12 |
| 3''-O-Acetyl polygalacin D | 0.45 | 0.03 |
| Platycodin A | 0.36 | 0.02 |
| Total | 100 | 5.62 |

The effect of β-glycosidase concentration on platycodin D production was investigated at various enzyme concentrations of 0.5 to 5 mg/mL using 5 mg/mL platycoside E in BFLE as the substrate for 3 h (Figure 5a). Platycodin D production increased with raising recombinant β-glycosidase concentration up to 3 mg/mL. However, the increase in platycodin D production was considerably reduced at concentrations higher than 3 mg/mL, indicating an optimal enzyme concentration of 3 mg/mL. Platycodin D production was carried out with 3 mg/mL enzyme for 3 h by varying the concentration of platycoside E in BFLE from 1 to 10 mg/mL (Figure 5b). Molar conversion yield decreased with increasing concentration of platycoside E in BFLE. However, platycodin D production constantly increased up to 5 mg/mL platycoside E in BFLE. At above 5 mg/mL platycoside E in BFLE, platycodin D production was considerably reduced. Thus, 5 mg/mL platycoside E in BFLE was selected as the substrate concentration for platycodin D production.

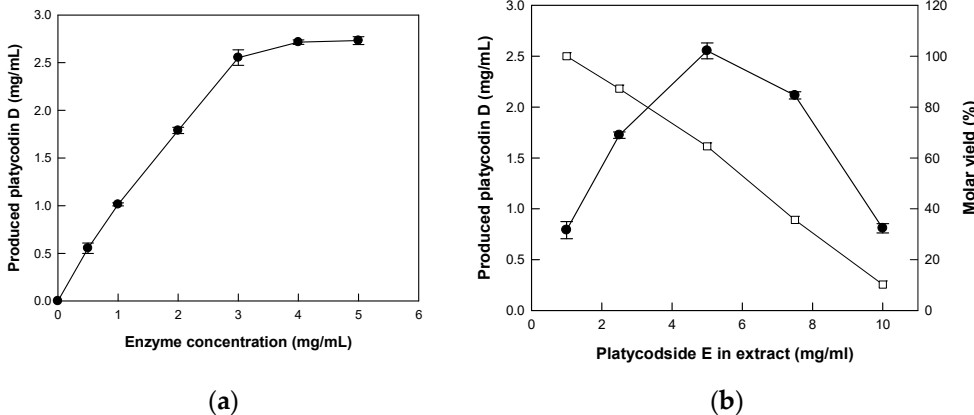

(**a**)                 (**b**)

**Figure 5.** Optimal enzyme and substrate concentrations for platycodin D production from BFLE by β-glycosidase from *C. owensensis*. (**a**) Optimal enzyme concentration. (**b**) Optimal substrate concentration. The concentration of produced platycodin D and molar conversion yield are represented by closed circles and open squares, respectively. Data are represented as means of triplicate experiments and the standard deviation was indicated by error bars.

### 2.5. Production of Platycodin D from BFLE by β-Glycosidase from C. owensensis

Under optimized reaction conditions, 3.95 mg/mL platycodin D was produced from 5 mg/mL platycoside E in BFLE by β-glycosidase from *C. owensensis* after 6 h and productivity and molar yield were represented to 658 mg/L/h and 100%, respectively. As a result, platycodin D concentration finally increased to 4.02 mg/mL from 0.07 mg/mL, the initial concentration in BFLE (Figure 6). The production was performed by the transformation pathway: platycoside E → platycodin D3 → platycodin D (Figure 1).

To date, platycodin D production has been carried out using PRE as substrate and enzymes such as cellulase [16], snailase [14], crude enzyme from *Cyberlindnera fabianii* [20], β-glucosidase from *A. usamii* [12] and β-glucosidase from *C. bescii* [17]. Of these, *C. bescii* β-glucosidase produced platycodin D with the highest concentration and productivity. Nevertheless, β-glycosidase from *C. owensensis* indicated 3.5- and 1.8-fold higher concentration and productivity of platycodin D, respectively, than β-glucosidase from *C. bescii*. These results would have been derived from the higher stability of β-glycosidase from *C. owensensis* at reaction temperature and the use of BFLE as an efficient substrate with high platycoside E content.

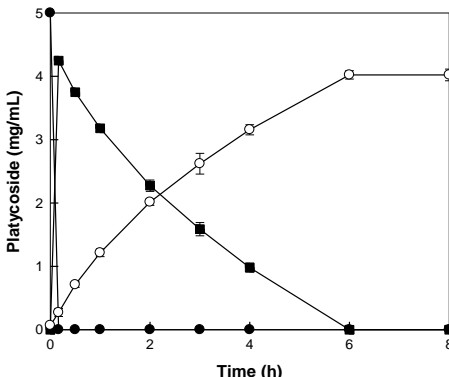

**Figure 6.** Time course reactions for platycodin D production from BFLE by β-glycosidase from *C. owensensis*. Closed circle, platycoside E; closed square, platycodin D3; and open circle, platycodin D. Data are represented as means of triplicate experiments and the standard deviation was indicated by error bars.

## 3. Materials and Methods

### 3.1. Bacterial Strains, Plasmid Vector and Gene Cloning

DNA temperate and expression vector for the β-glycosidase gene (GenBank accession no. ADQ03897) cloning and host cells of enzyme expression were *C. owensensis* DSM 13100 (DSMZ, Braunschweig, Germany), and pET-29b (+) vector (Novagen, Darmstadt, Germany), and *E. coli* BL21 strain, respectively. The β-glycosidase gene from *C. owensensis* was cloned by the one-step isothermal DNA assembly method. The DNA fragments of the β-glycosidase gene and the pET-29b (+) expression vector were amplified by PCR with the following primers: forward primer for β-glycosidase gene, 5′-AGCAGCGAAAACCTGTATTTTCAGGGACATATGAG TTTTCCAAAAGGATTTTTGTGGGGT-3′; reverse primer for β-glycosidase gene, 5′-ATCTCAGTGGTGGTGGTGGTGGTG CTCGAGTTATGAATTTTCCTTTATATA CTGCTGATA-3′; forward primer for expression vector, 5′-TATCAGCAGTATATAAAGGAAAATTCATAACTCGAGCACCACCACCACCACCACTGAGAT-3′; and reverse primer for expression vector, 5′-ACCCCACAAAAATCCTTTTGGAAAACTCATATGT CCCTGAAAATACAGGTTTTCGCTGCT-3′. The amplified β-glycosidase gene from *C. owensensis* and pET-29b (+) expression vector were synthesized by Phusion High-Fidelity DNA polymerase (ThermoFisher Scientific, Waltham, MA, USA) using PCR. And the synthesized DNA fragments were ligated using Master Mix for Gibson Assembly (New England Biolabs, Ipswich, MA, USA) [21] and the ligated DNA was transformed into *E. coli* BL21.

### 3.2. Bacteria Culture for Enzyme Expression

The recombinant *E. coli* BL 21 for expression of β-glycosidase from *C. owensensis* were cultivated in a 2000 mL baffled glass Erlenmeyer flask containing 450 mL Luria-Bertani medium mixed with 50 μg/mL antibiotic (kanamycin) at 37 °C with 200 rpm agitation on a shaking incubator. When the optical density reached 0.6 (at 600 nm) of recombinant *E. coli* BL21 growth, 100 mM IPTG was added for expression of β-glycosidase from *C. owensensis* and the cells were then incubated at 16 °C for another 14 h.

### 3.3. Preparation of β-Glycosidase from C. Owensensis

The cultured recombinant *E. coli* BL 21 cells were harvested by centrifugation, suspended in Ni-NTA lysis buffer with 1 mg/mL lysozyme and disrupted using a ultrasonicator (KBT, Sungnam, Korea) on ice for 20 min. Cell debris and undisrupted cells were removed by centrifugation, and supernatant was applied to a HisTrap HP affinity column on a Bio-Rad Profinia™ purification system (Bio-Rad, Hercules, CA, USA). The collected fractions with hydrolytic activity were dialyzed against 50 mM citrate phosphate buffer (CPB) pH 5.5 for 16 h at 4 °C. The dialyzed solution was used as a purified β-glycosidase from *C. owensensis*. The concentration of purified enzyme was measured by Bradford protein assay.

### 3.4. SDS-PAGE and Gel-Filtration Chromatography

The subunit molecular mass of the recombinant β-glycosidase from *C. owensensis* was examined by SDS-PAGE (Bio-Rad, Hercules, CA, USA). Cellular debris, crude enzyme extract, and purified β-glycosidase were used with 1 mg/mL of concentration to compare the degree of purification. Coomassie blue was used as a dye for visualization of all protein bands at each purification step. Gel-filtration chromatography was used for determination of molecular mass of the recombinant β-glycosidase from *C. owensensis*. The recombinant β-glycosidase was transferred through a HiPrep™ 16/60 Sephacryl® S-300 HR column (GE Healthcare, Chicago, IL, USA) and then the enzyme-bounded column was eluted with 50 mM CPB (pH 5.5) containing 150 mM NaCl at a flow rate of 0.5 mL/min. The cultured recombinant *E. coli* BL 21 cells were harvested by centrifugation, suspended in Ni-NTA lysis buffer (50 mM sodium/phosphate buffer containing 5 mM imidazole and 300 mM NaCl, pH 7.0) with 1 mg/mL lysozyme. The purified enzyme was eluted with the reference proteins: ovalbumin,

conalbumin, aldolase and catalase have molecular masses of 43, 75, 158, and 232 kDa, respectively. The retention time of β-glycosidase from *C. owensensis* was computed by comparison with the reference protein migration lengths.

### 3.5. Preparation of BFLE

Two-year-old balloon flower leaf cultivated in Bonghwa-gun was used to prepare the extract. The balloon flower leaf was dried for 72 h at 40 °C in a dry oven. The dried balloon flower leaves were ground using an electric grinder. The 100 g of fine balloon flower leaf powder was extracted using 1 L of 80% (*v/v*) methanol at 70 °C for 24 h. The balloon flower leaf extract was filtered, the methanol was completely removed by rotary evaporator and the residue was dissolved in the same volume of distilled water. To prevent a Maillard reaction of free sugars with the β-glycosidase from *C. owensensis* at enzyme reaction temperature above 70 °C, the sugars in the extract were removed using Diaion HP20 resin (Sigma-Aldrich, St. Louis, MO, USA) column. The extract was loaded onto the column, and then the platycoside-adsorbed resin was washed with distilled water. After washing, the column absorbed platycosides eluted with methanol at a flow rate of 0.5 mL/min. The methanol in the eluent was evaporated and 1 L of distilled water was added into residue. The resulting BFLE was used for platycodin D production.

### 3.6. Hydrolytic Activity Assay

The enzyme reaction was performed at 80 °C in 50 mM CPB (pH 5.0) containing 50 μg/mL β-glycosidase from *C. owensensis* and 400 μg/mL platycoside for 10 min. The specific activity of β-glycosidase from *C. owensensis* for platycosides including deapi-platycodin D, deapi-platycodin D3, deapi-platycoside E, platycodin D, platycodin D3 and platycoside E was evaluated with various concentrations of the enzyme (5–50 μg/mL) and 400 μg/mL of each platycoside for 10 min. The effects of temperature and pH on the activity of β-glycosidase from *C. owensensis* for PE were examined with 1 mM pNP-β-D-glucopyranoside by varying the temperature from 65 to 95 °C at a pH of 5.0 for 10 min and by varying the pH from 4.0 to 7.5 at 75 °C for 10 min, respectively. The effect of thermostability of the β-glycosidase from *C. owensensis* was monitored as a function of incubation time by maintaining the solution of enzymes at 70, 75, 80, 85 and 90 °C in 50 mM CPB (pH 5.0). After incubating, the reaction samples were assayed with 1 mM pNP-β-D-glucopyranoside in 50 mM CPB (pH 5.0) at 80 °C for 10 min. Samples were withdrawn at regular time intervals and assayed.

### 3.7. Biotransformation of Platycoside

The optimal concentration of β-glycosidase for platycodin D production from BFLE was determined by varying the concentration of β-glycosidase from 0.5 to 5 mg/mL with 5 mg/mL platycoside E in BFLE. The optimal concentration of platycoside E in BFLE as a substrate was determined by varying the concentration of platycoside E from 1 to 10 mg/mL at a constant enzyme concentration of 3 mg/mL. The reactions were performed in 50 mM CPB (pH 5.0) for 3 h at 80 °C. The time-course reaction for converting platycoside E in BFLE to platycodin D was performed in 50 mM CPB (pH 5.0) with 3 mg/mL of β-glycosidase and 5 mg/mL platycoside E in BFLE (containing 5 mg/mL platycoside E (PE) and 0.07 mg/mL platycodin D (PD)) at 80 °C for 8 h.

### 3.8. HPLC Analysis

After enzyme reaction, the reaction solution was extracted by the same volume of n-butanol with an internal standard (digoxin). The n-butanol fraction in the extracted solution was evaporated until it was completely dried and the same volume of methanol was added. Platycoside analysis was performed using an Agilent 1100 series HPLC system at 203 nm with a hydrosphere C18 column (4.6 × 150 mm; YMC, Kyoto, Japan). The column was eluted at 37 °C for 60 min at a flow rate of 1 mL/min with the following gradient of acetonitrile/water (*v/v*): from 10/90 to 40/60 for 30 min; from 40/60 to

90/10 for 15 min; from 90/10 to 10/90 for 5 min; and constant at 10/90 for 10 min. All of the platycoside standards were purchased from Ambo Laboratories (Daejeon, Korea)

## 4. Conclusions

In this study, the substrate specificity of β-glycosides from *C. owensensis* was measured using aryl-glycosides and platycosides, and the enzyme reaction conditions for the production of platycodin D were optimized. Under the optimum enzyme reaction conditions, the recombinant β-glycosides from *C. owensensis* completely converted BFLE platycoside E into platycodin D with the highest concentration and productivity reported so far. To the best of our knowledge, this is the first time BFLE was used as an efficient substrate by recycling leaves discarded from Platycodi radix production. Our results greatly improve the production process for deglycosylated platycosides.

**Author Contributions:** Y.-S.K. and S.-J.Y. supervised this study. K.-C.S. performed most of the experiments and organized the results. M.-J.S. carried out gene cloning and enzyme purification. K.-C.S., D.-W.K., S.-J.Y., and Y.-S.K. designed the experiments and contributed to the data analysis. K.-C.S., M.-J.S., D.-W.K., S.-J.Y., and Y.-S.K. wrote the manuscript.

**Funding:** This study was supported by the Individual Basic Science & Engineering Research Program through the National Research Foundation grant funded by the Ministry of Science and ICT, Korea (NRF-2017R1D1A1B03033762).

**Conflicts of Interest:** The authors declare no conflict of interest.

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
