# Peer review of "Characterization of β-Glycosidase from Caldicellulosiruptor owensensis and Its Application in the Production of Platycodin D from Balloon Flower Leaf"

_catalysts, doi:10.3390/catal9121025_

Round 1
Reviewer 1 Report
I have thoroughly analyzed the manuscript, and I think it should be published in Catalysts after minor corrections.
The only part requiring correction is Conclusions, and the authors should rewrite it.
authors should delete the first sentence:
In this study, the biochemical properties of β-glycosides from C. owensensis were characterized,
and add information:
The substrate specificity of β-glycosidase from C. owensensis was measured using aryl-glycosides and platycosides.
Please transfer the graphs from Supplementary Materials section to the main part of the manuscript.
Author Response
Response to Reviewer #1
Point 1: The only part requiring correction is Conclusions, and the authors should rewrite it. Authors should delete the first sentence: In this study, the biochemical properties of β-glycosides from C. owensensis were characterized, and add information: The substrate specificity of β-glycosidase from C. owensensis was measured using aryl-glycosides and platycosides.
Response 1: Thank you for your suggestion. As you suggested, the sentence was revised to “In this study, the substrate specificity of β-glycosidase from C. owensensis was measured using aryl-glycosides and platycosides, and the enzyme reaction conditions for the production of platycodin D were optimized.” (Line 262-264 of the revised manuscript)
Point 2: Please transfer the graphs from Supplementary Materials section to the main part of manuscript.
Response 2: Thank you for your good comment. The graphs was transferred from Supplementary Materials section to main part in the revised manuscript. (Figure 3 of the revised manuscript)

Reviewer 2 Report
Dear Author
Generally speaking, I find your manuscript well written and interesting. Although, for some years now I have been involved in research related to carbohydrate active enzymes, my main field of application is the lignocellulose deconstruction. Therefore, I might not have extensive knowledge and/or an overview of your specific topic.
Major comments:
I would slightly improve the overall structure of this manuscript. I have a feeling that methods are largely mixed with the results and the discussion section. Some of them are not even described in the methods section, while the results are shown in the main results section, or the reverse. For example, the figure captures contains an extensive description of the method used, like Fig.2 where the protocol to perform the gel-filtration chromatography is given. Instead, it should be described in the methods section, and currently it is not. Similar changes could be made to improve captions of Fig.3, 4 and 5. How was the concentration of the enzyme calculated? According to the shown SDS-PAGE (Fig 2a), the authors did not manage to well purify the protein from the cell lysate? How in this case have you calculated the concentration of your enzyme? Is it really quantifiable? In general there is very little interesting discussion given in the results and discussion section. While, I also prefer avoiding too much speculation and extensive discussions in general, this part of your manuscript could be slightly improved. You do compare the activity and efficiency of your protein to the previously reported studies, which is correct. But again, at least when it comes to the efficiency I am not sure how well it can be compared in your experimental conditions. By contrast, you could add some additional discussion, e.g. lane 108, where you say that the enzyme did not cleave the residues of arabinose etc..any explanation why? Please revise accordingly the whole section. Table 2 is well described in the main text, lines 126-129, so in principle could be avoided.
Minor comments:
Perhaps provide a reference to the Fig.1. I guess this deglycosilation pathways have been previously deciphered by somebody else? Is there any reason behind using both pNP- and oNP-labelled substrates? Rephrase some sentences for better clarity, e.g. lines 82-85; lines 132-134 Line 32 “significant” refers to what? Line 96, I would change “for a diversity of time points” to something like “different time periods/points” or similar. Line 160 “..and productivity..” change to “and showed productivity..” The sentence in lines 154-156, while confirming what is known about the deglycosilation pathway of platycoside E, is quite speculative here, as it was not really evidenced by the experimental design. Please correct me if I am wrong. Line 199, I guess it is the supernatant that was applied to the affinity purification? Please rewrite for clarity.
Author Response
Response to Reviewer #2
Point 1: I would slightly improve the overall structure of this manuscript. I have a feeling that methods are largely mixed with the results and the discussion section. Some of them are not even described in the methods section, while the results are shown in the main results section, or the reverse. For example, the figure captures contains an extensive description of the method used, like Fig.2 where the protocol to perform the gel-filtration chromatography is given. Instead, it should be described in the methods section, and currently it is not. Similar changes could be made to improve captions of Fig.3, 4 and 5.
Response 1: Thank you for your pointing out. As you mentioned, captions of figures were revised to reduce description of the methods. (Line 71−76, line 90−92, line 93−96, line 146−150, and line 166−168 of the revised manuscript) Some of the captions were moved to the Materials and methods section. (Line 240−242 and line 250−251 of the revised manuscript)
Also, the sentences for protocol to perform the gel-filtration chromatography were newly added as “SDS-PAGE and gel-filtration chromatography” section in Materials and methods. (Line 203−216 of the revised manuscript)
Point 2: How was the concentration of the enzyme calculated? According to the shown SDS-PAGE (Fig 2a), the authors did not manage to well purify the protein from the cell lysate? How in this case have you calculated the concentration of your enzyme? Is it really quantifiable?
Response 2: Thank you for your suggestion. The concentration of the enzyme was calculated by Bradford protein assay. It was not described in the original manuscript because it is a common method for most measurements of enzyme concentration. However, for better understanding, we newly inserted this information in the revised manuscript. (Line 200−201 of the revised manuscript).
Lane 4 in Fig. 2 represents the purified enzyme. Compared to crude enzyme extract (lane 3), you can see the remarkably purified enzyme at 53 kDa. We have already described information for purification fold and yield in the original manuscript. (Line 62−63 of the original manuscript) Although other bands appeared blurry on SDS-PAGE, they were not a problematic level of purification. In addition, no peaks of other proteins were found in the results of gel-filtration chromatography. For better understanding, we added the sentence “No peak other than β-glycosidase was detected in the chromatography (data not shown).” (Line 70 of the revised manuscript)
Therefore, the enzyme concentration was calculated by protein assay with the fraction of lane 4. This is a level that is sufficiently quantitative.
Point 3: In general there is very little interesting discussion given in the results and discussion section. While, I also prefer avoiding too much speculation and extensive discussions in general, this part of your manuscript could be slightly improved. You do compare the activity and efficiency of your protein to the previously reported studies, which is correct. But again, at least when it comes to the efficiency I am not sure how well it can be compared in your experimental conditions. By contrast, you could add some additional discussion, e.g. lane 108, where you say that the enzyme did not cleave the residues of arabinose etc..any explanation why? Please revise accordingly the whole section. Perhaps provide a reference to the Fig.1. I guess this deglycosilation pathways have been previously deciphered by somebody else.
Response 3: Thank you for your comment. Since there is not much research on glycosidase for hydrolysis of platycosides, it was difficult for the enzyme used in this study to compare exactly with previously reported studies. Therefore, only activity comparisons with β-glucosidase from C. bescii, which has a similar hydrolytic pathway, were discussed in the original manuscript. (Line 113−115 of the original manuscript)
However, we added additional discussions in the revised manuscript as you suggested to improve the discussion section of the manuscript. (Line 105−109 of the revised manuscript)
Point 4: Is there any reason behind using both pNP- and oNP-labelled substrates?
Response 4: Thank you for your concern. In general, substrate specificity for aryl-glycosides is performed to characterize glycosidase. In this study, aryl-glycosides were used to characterize glycosidase from C. owensensis and to correlate substrate specificity for platycosides.
Point 5: Rephrase some sentences for better clarity, e.g. lines 82-85; lines 132-134 Line 32 “significant” refers to what?
Response 5: Thank you for your pointing out. Line 82-85 in the original manuscript was revised as follows: “The recombinant β-glycosidase from C. owensensis showed a maximum hydrolytic activity at pH 5.0 and 80 °C (Figure 3). Optimum pH and temperature of recombinant enzymes used for platycoside deglycosylation, β-glucosidases from Caldicellulosiruptor bescii [17] and Dictyoglomus turgidum [18], were pH 5.5 and 80 °C, and pH 6.5 and 80 °C, respectively.” (Line 78−81 of the revised manuscript).
Line 132-134 in the original manuscript was revised as follows: “The effect of β-glycosidase concentration on platycodin D production was investigated at various enzyme concentrations of 0.5 to 5 mg/ml using 5 mg/ml platycoside E in BFLE as the substrate for 3 h (Figure 4a).” (Line 135−137 of the revised manuscript)
To avoid confusion, line 32 “significant” in the original manuscript was revised to “major” in the revised manuscript. (Line 32 of the revised manuscript)
Point 6: Line 96, I would change “for a diversity of time points” to something like “different time periods/points” or similar.
Response 6: Thank you for your suggestion. As you suggested, the phrase of “a diversity of time periods” was revised to “different time periods”. (Line 95 of the revised manuscript)
Point 7: Line 160 “..and productivity..” change to “and showed productivity..”
Response 7: Thank you for your pointing out. The sentence was revised to “Of these, β-glucosidase from C. bescii produced platycodin D with the highest concentration and productivity”. (Line 160−161 of the revised manuscript)
Point 8: The sentence in lines 154-156, while confirming what is known about the deglycosilation pathway of platycoside E, is quite speculative here, as it was not really evidenced by the experimental design. Please correct me if I am wrong.
Response 8: Thank you for your concern. In Fig 6. of the revised manuscript, you can see that platycodin D3 increases as platycoside decrreases, and platycodin D increases as platycodin D3 decreases. These results indicate deglycosylation pathway. For a better understanding, new sentence was added in the substrate specificity section of the revised manuscript as follows: “Platycoside E, deapi-platycoside E, platycodin D3, and deapi-platycodin D3 were converted to platycodin D3, deapi-platycodin D3, platycodin D, and deapi-platycodin D, respectively.” (Line 112−113 of the revised manuscript)
Point 9: Line 199, I guess it is the supernatant that was applied to the affinity purification? Please rewrite for clarity.
Response 9: Thank you for your pointing out. As you mentioned, the sentence was revised to “Cell debris and undisrupted cells were removed by centrifugation, and supernatant was applied to a HisTrap HP affinity column equilibrated with 50 mM sodium/phosphate buffer (pH 7.0) on a Bio-Rad ProfiniaTM purification system.” (Line 196−198 of the revised manuscript)

Round 2
Reviewer 2 Report
Dear Authors
Thank you very much for your work in addressing my comments. I find this manuscript improved, even though I think it could still lack scientific rigour in some cases. For example, in my opinion the enzyme was not well purified, and the contamination of co-purified proteins is still quite significant. In most of the cases it does not prevent to characterise its specific activity etc. However, I would not express this activity per mg of purified protein, if your protein is not really pure.